 **eLIFE**

# Essential role of lncRNA binding for WDR5 maintenance of active chromatin and embryonic stem cell pluripotency

Yul W Yang[1,2†], Ryan A Flynn[1,2†], Yong Chen[3,4], Kun Qu[1,2], Bingbing Wan[3,4], Kevin C Wang[1,2], Ming Lei[3,4], Howard Y Chang[1,2]*

[1]Howard Hughes Medical Institute, Stanford University School of Medicine, Stanford, United States; [2]Program in Epithelial Biology, Stanford University School of Medicine, Stanford, United States; [3]Department of Biological Chemistry, University of Michigan, Ann Arbor, United States; [4]Howard Hughes Medical Institute, University of Michigan, Ann Arbor, United States

**Abstract** The WDR5 subunit of the MLL complex enforces active chromatin and can bind RNA; the relationship between these two activities is unclear. Here we identify a RNA binding pocket on WDR5, and discover a WDR5 mutant (F266A) that selectively abrogates RNA binding without affecting MLL complex assembly or catalytic activity. Complementation in ESCs shows that WDR5 F266A mutant is unable to accumulate on chromatin, and is defective in gene activation, maintenance of histone H3 lysine 4 trimethylation, and ESC self renewal. We identify a family of ESC messenger and lncRNAs that interact with wild type WDR5 but not F266A mutant, including several lncRNAs known to be important for ESC gene expression. These results suggest that specific RNAs are integral inputs into the WDR5-MLL complex for maintenance of the active chromatin state and embryonic stem cell fates.

*For correspondence:
howchang@stanford.edu

†These authors contributed equally to this work

Competing interests: The authors declare that no competing interests exist.

## Introduction

An orchestra of chromatin readers, writers, and erasers act on diverse covalent histone modifications to establish particular cell fates (*Rando and Chang, 2009*). By maintaining such histone modifications through cell divisions, the cell state can then be epigenetically transferred from generation to generation, eventually establishing tissues and complex organisms. Histones are modified by large protein complexes with multiple coenzymes that modulate and regulate catalytic activity, thus allowing fine regulation of histone marks.

Histone H3 lysine 4 trimethylation (H3K4me3) is a mark of transcriptionally active chromatin, generated through the catalytic activity of the MLL family of proteins. The MLL family of H3K4 methylases are evolutionarily conserved (known as COMPASS in yeast and trithorax in *Drosophila*) (*Smith et al., 2011*), and *MLL* translocations are an important cause of human leukemias (*Meyer et al., 2009*). Members of the MLL family encode single SET domain-containing H3K4 methylases, and are associated with WDR5, Ash2L, RbBP5, and additional proteins to regulate activity. Of the various cofactors that regulate MLL activity, WDR5 is a particularly important multifunctional adaptor protein that can discriminate posttranslational modifications on histone tails, as well as bind to the MLL complex to regulate gene activation (*Wysocka et al., 2005*; *Migliori et al., 2012*). WDR5 is particularly important for mammalian embryonic stem cell (ESC) self renewal and maintenance of active chromatin for pluripotency genes, and WDR5 is required for efficient generation of induced pluripotent stem cells from differentiated somatic cells (*Ang et al., 2011*; *Li et al., 2012*).

WDR5 has recently been shown to bind individual long noncoding RNAs (lncRNAs) (*Wang et al., 2011*; *Gomez et al., 2013*). LncRNAs are capped, spliced, polyadenylated RNA transcripts ranging

**eLife digest** If all the DNA contained within a single human cell were stretched out it would be about three meters long. To fit this length of DNA into the nucleus of the cell, it is packaged into a compact structure called chromatin. If a cell wants to express one of the genes in the DNA in order to produce a protein, it must unpack part of the chromatin to give an enzyme called RNA polymerase access to the DNA to produce messenger RNA. However, other enzymes—often working with other enzymes, proteins and molecules called cofactors—can modify the structure of the chromatin in a way that leads to changes in the expression of nearby genes.

The protein WDR5 binds to enzymes and helps to modify chromatin so that nearby genes can be 'switched on'. However, WDR5 also binds to RNA molecules that are not involved in the expression of genes as proteins. These long non-coding RNA molecules (lncRNAs) are thought to act as scaffolds that guide the WDR5-enzyme complex to specific stretches of DNA. However, it is not clear if lncRNAs might be performing roles that affect the function of these enzyme complexes directly—as has been observed for other cofactors.

Yang, Flynn et al. have now investigated the role of lncRNA molecules further, and identified a pocket on the structure of WDR5 where these molecules bind. A mutation in this pocket that blocks binding to lncRNA did not stop purified WDR5 protein from forming active enzyme complexes when tested in the lab. However, it seems that WDR5 needs to bind to the lncRNAs to function properly inside living cells. WDR5 binds to hundreds of lncRNAs within a cell, and the mutation that blocks this binding reduced both the amount of time that WDR5 protein survives in a cell and the amount of time it spends bound to chromatin. As such, this mutation seems to reduce the ability of the WDR5-enzyme complex to activate chromatin and 'switch on' genes.

WDR5 is also known to be important in mammalian embryonic stem cells—cells that have the potential to become all the different types of cells found in the body. Yang, Flynn et al. uncovered that the mutations in WDR5 that abolish lncRNA binding also affect the expression of genes that help mouse stem cells to maintain this potential. Since different cell types have distinct patterns of active chromatin, the next challenge will be to understand whether different lncRNAs bind to WDR5 to switch on unique set of genes in each cell type.

from several hundred to kilobases in length (*Derrien et al., 2012*; *Rinn and Chang, 2012*). Specific lncRNAs bind repressive or activating chromatin modification complexes, and localize these activities to specific gene loci (reviewed by *Wang and Chang (2011)*). For example, the lncRNA XIST binds the Polycomb Repressive Complex 2 (PRC2) to cause histone H3 lysine 27 trimethylation and silence the X chromosome for dosage compensation in females (*Morey and Avner, 2011*). As another example, the lncRNA HOTAIR acts as a molecular scaffold, binding both PRC2 and the H3K4 demethylase LSD1 complex to silence hundreds of loci throughout the genome (*Rinn et al., 2007*; *Gupta et al., 2010*; *Tsai et al., 2010*; *Chu et al., 2011*). Additional lncRNAs can bind to messenger RNAs to control their decay via interaction with the Staufen 1 protein (*Gong and Maquat, 2011*; *Kretz et al., 2013*).

In contrast, several lncRNAs bind to WDR5 to facilitate H3K4me3 and epigenetic activation. HOTTIP is an enhancer-like lncRNA of the human *HOXA* locus that coordinates expression of *HOXA9* to *HOXA13*, which are important for distal identity (*Wang et al., 2011*). HOTTIP RNA directly binds to the WDR5 protein to recruit the MLL H3K4 methylase complex to maintain H3K4me3. In addition, recent research has shown that the NeST lncRNA also binds WDR5 to upregulate IFN-γ expression through H3K4me3 (*Gomez et al., 2013*), suggesting the existence of multiple different enhancing lncRNAs that function via WDR5 interactions. Many enhancers and promoters often produce lncRNAs, and some of these lncRNAs have enhancer-like functions in the activation of nearby genes by binding the Mediator complex (*Kim et al., 2010*; *Orom et al., 2010*; *Hung et al., 2011*; *Rada-Iglesias et al., 2011*; *Lai et al., 2013*). Currently, it is believed that lncRNAs function by guiding localization of chromatin modification machinery as simple scaffolds. However, like coenzymes that modulate protein activity, it is possible that lncRNAs may affect protein function more directly.

To examine the roles of RNA binding on WDR5 functionality, here we identify a RNA binding site on WDR5, and characterized its function in vitro and in vivo. Inability to bind RNAs does not affect MLL

complex formation or catalytic activity in vitro. However, ability to bind RNAs is crucial for WDR5 function in cells. We find that WDR5 binds over a thousand endogenous RNAs and that RNA binding is essential for WDR5 maintenance of ESC pluripotency. Inability to bind RNAs greatly reduced duration of WDR5 protein and its occupancy on chromatin, impaired global H3K4me3 levels, and reduced expression of genes required for embryonic stem cell state. Overall, these results suggest an unexpected role for RNA binding in regulating the lifespan of epigenetic fates.

## Results

### Discovering the lncRNA binding interface on WDR5

To determine the lncRNA binding site, we performed alanine scanning mutagenesis of WDR5, guided by its crystal structure (*Trievel and Shilatifard, 2009*; *Odho et al., 2010*; *Avdic et al., 2011*) (*Figure 1A,B*). WDR5 is a barrel-shaped protein with several charged clefts on its surface, several of which are known to mediate protein–protein interactions with MLL, histone H3 tail, or RbBP5 (*Figure 1A*, *Figure 1—figure supplement 1A,B*). We generated 19 WDR5 point mutants, expressed them in *Escherichia coli* as GST-fusion proteins, and purified them to homogeneity (*Figure 1—figure supplement 1C*). Four out of 19 mutants significantly reduced the ability to retrieve HOTTIP lncRNA in vitro: Y228A, L240A, K250A, and F266A. These WDR5 mutations defined a cleft between blades 5 and 6, partially encompassing the same surface previously described to bind RbBP5 amino acids 371–381 (*Odho et al., 2010*; *Avdic et al., 2011*). Thus, a focal binding site defines the interaction between WDR5 and HOTTIP. To confirm that HOTTIP and RbBP5 bind to the same or overlapping sites on WDR5, we pre-incubated wild type GST-WDR5 with an excess of RbBP5 peptide (amino acids 371–381) or control H3K4me3 peptide (amino acids 1–20), and then assayed for HOTTIP binding (*Figure 1C*). Whereas addition of H3K4me3 peptide had no effect, pre-incubation with RbBP5 peptide prevented HOTTIP binding to WDR5, thus confirming the shared binding cleft.

To verify the lncRNA binding site in living cells, we conducted in vivo RNA immunoprecipitation (RIP) experiments with select WDR5 mutants in 293T cells (*Figure 1D*). Whereas the D107A and R181A mutations caused little effect on RbBP5 binding compared with wild type (~95%), the K250 mutation reduced RbBP5 binding (~62.5%) as previously described (*Odho et al., 2010*; *Avdic et al., 2011*). Furthermore, the F266A mutation actually increased RbBP5 binding (~120%), suggesting that loss of binding to HOTTIP increases the ability to bind RbBP5. Consistent with the direct in vitro binding assay, both K250A and F266A mutations fully abrogated WDR5 binding to HOTTIP in vivo. In contrast, mutations at D107A and R181A showed minimal effects on WDR5-HOTTIP interactions. K250A and F266A did not affect the accumulation of HOTTIP lncRNA (*Figure 1—figure supplement 1D*), and both negative controls U1 and HOTAIR did not enrich binding to WDR5 (*Figure 1—figure supplement 1E*). Taken together, the in vitro and in vivo analyses demonstrate that HOTTIP RNA binds WDR5 through a specific binding pocket.

### A selective mutant reveals the importance of lncRNA binding on WDR5-mediated gene activation

To pinpoint the functional consequences of a selective lncRNA-binding mutation of WDR5, we further analyzed WDR5 F266A. In contrast to the other HOTTIP binding mutations, WDR5 F266A is defective in lncRNA binding in vitro and in vivo, but without any defects in binding MLL complex subunits RbBP5 or MLL1 in immunoprecipitation experiments (*Figure 1D*). We reasoned that the F266A mutation offered an experimental strategy to isolate the requirement of lncRNA binding for WDR5 function.

To confirm the lncRNA selectivity of the F266A mutation, we examined its effects on MLL complex structure and catalytic activity in vitro. As seen by isothermal calorimetry, the F266A mutation minimally affects the affinity of WDR5 for RbBP5, MLL1, or H3 peptides, in contrast to the Y228A mutation (RbBP5 binding defective) or D107A mutation (MLL1 and H3 binding defective) (*Figure 2A*). This was further confirmed using GST protein pull down assays, in which GST-WDR5 F266A displays no deficits in binding RbBP5, Ash2L, and MLL1 purified proteins (*Figure 2B*). Finally, in contrast to the MLL1/H3 or RbBP5 binding mutants, the F266A mutation did not significantly decrease in vitro MLL1 complex methylase activity compared to wild type (*Figure 2C*). Thus, the F266A mutation appears to singularly affect lncRNA binding, without altering MLL complex structure or catalytic activity.

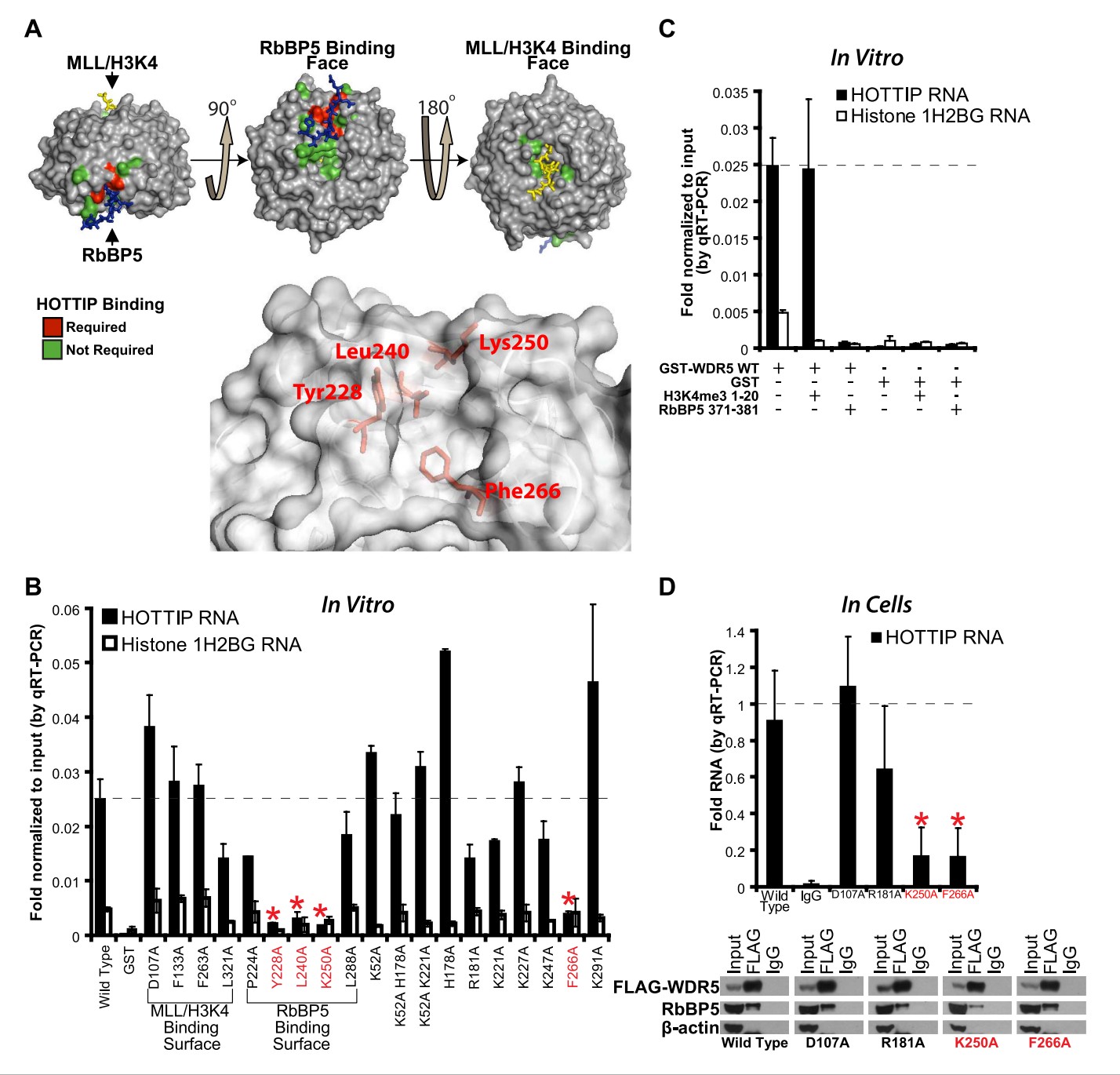

**Figure 1**. HOTTIP lncRNA binding surface overlaps with the RbBP5 binding surface on WDR5. (**A**) Crystal structure of WDR5 (PDB:3P4F, visualized with PyMol) reveal that mutations abrogating HOTTIP RNA binding align along a cleft between blades 5 and 6, opposite the H3K4/MLL binding site. This binding surface overlaps the site for RbBP5. Top: schematic of tested and HOTTIP binding mutations. Bottom: magnification of HOTTIP binding cleft. Yellow: MLL peptide. Blue: RbBP5 peptide. (**B**) qRT-PCR results of indicated GST-WDR5 mutants tested by in vitro assay for binding to HOTTIP RNA or control Histone 1H2BG RNA. (**C**) RbBP5 peptide competition assay indicates that RbBP5 amino acids 371–381 can fully inhibit HOTTIP binding to WDR5. Wild type data same as in *Figure 1B*. (**D**) Average fold pulldown of full length HOTTIP by cell-based native RNA immunoprecipitation of select WDR5 mutants. Top: qRT-PCR results of RNA immunoprecipitation of FLAG-WDR5 mutants. All values are normalized to input, then to FLAG pulldown of wild type and positive control D107A. Negative RNA controls (U1, HOTAIR) and reaction without reverse transcription (-RT) show minimal enrichment (*Figure 1—figure supplement 1*, data not shown). Bottom: representative western blots of immunoprecipitations.

The following figure supplements are available for figure 1:

**Figure supplement 1**. lncRNA binding surface of WDR5 and WDR5 RIP-qRT-PCR standards.

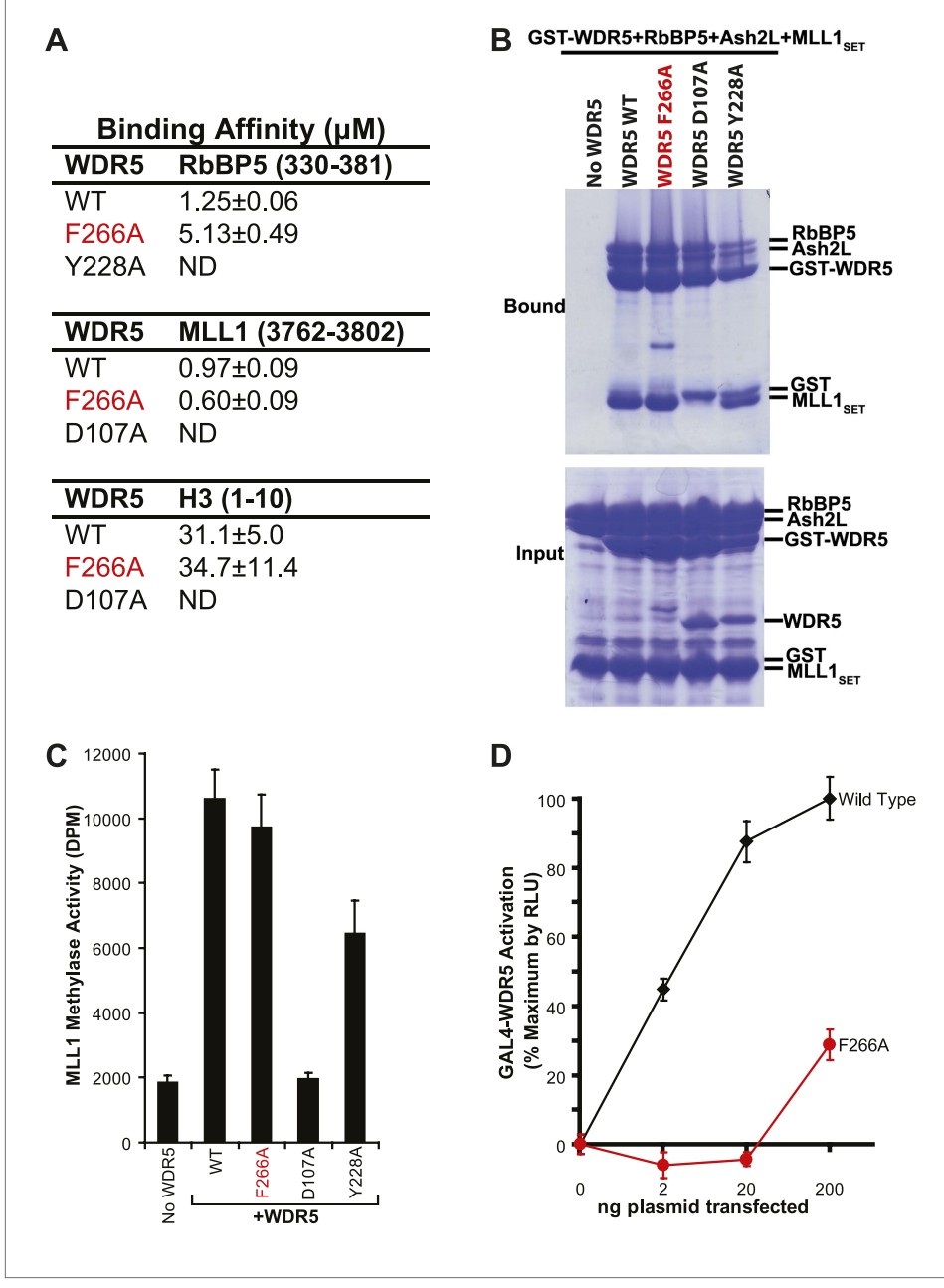

**Figure 2**. WDR5 F266A lncRNA binding mutant does not affect MLL complex formation or catalytic function, but shows decreased ability to activate target genes in 293T cells. (**A**) WDR5 binding affinity to RbBP5, MLL1, or H3 peptides by isothermal calorimetry. ND, not detectable. (**B**) WDR5 F266A does not have decreased binding to RbBP5, Ash2L and MLL1SET domain proteins in GST protein pull down assays. (**C**) WDR5 F266A does not affect histone methylase activity of the MLL1 complex. (**D**) GAL4-WDR5 F266 is defective in activating luciferase expression, as seen in luciferase titration tests.

   To test the effects of the F266A mutation on gene activation in cells, we transfected increasing amounts of plasmids encoding GAL4-WDR5 WT or GAL4-WDR5 F266A to activate luciferase reporter gene expression (*Figure 2D*), as previously described (*Wysocka et al., 2005*). Surprisingly, WDR5 F266A demonstrated a severe defect in luciferase gene activation, requiring >100-fold transfected plasmid compared with wild type to achieve similar reporter gene activity. Thus, even though the WDR5 F266A mutation can bind the full MLL complex and does not affect catalytic activity, the inability to bind lncRNAs strongly compromises WDR5 function in activating gene expression by reporter gene assay.

## lncRNA binding to WDR5 is essential for protein stability and H3K4 trimethylation in embryonic stem cells

WDR5 has been recently shown to be required to maintain H3K4 trimethylation (H3K4me3) in mouse embryonic stem cell (ESC) genes for pluripotency and self renewal (*Ang et al., 2011*). To test whether WDR5 requires lncRNA binding for physiologic activity, we created 'rescue' complementation ESC lines that replace endogenous mouse WDR5 with either wild type human WDR5 or F266A mutant WDR5 (*Figure 3A*). ESCs were infected by lentiviruses containing two tandem gene expression cassettes (*Ang et al., 2011*). The first cassette constitutively expresses a highly efficient shRNA to repress endogenous mouse WDR5. WDR5 deficiency is rescued by the second cassette, in which human WDR5 wild type (WT) or human WDR5 F266A linked to GFP is expressed under control of a doxycycline (dox)-inducible promoter. In the presence of dox, the ability of WDR5 mutant to support ESC self renewal can be compared; upon dox withdrawal, the half-life of the mutant protein and its regulatory impact are further revealed.

GFP+ cells were sorted using consistent fluorescence parameters, cultured for 4 days in the presence (+dox) and absence (−dox) of doxycycline, and then analyzed by western blot and qRT-PCR.

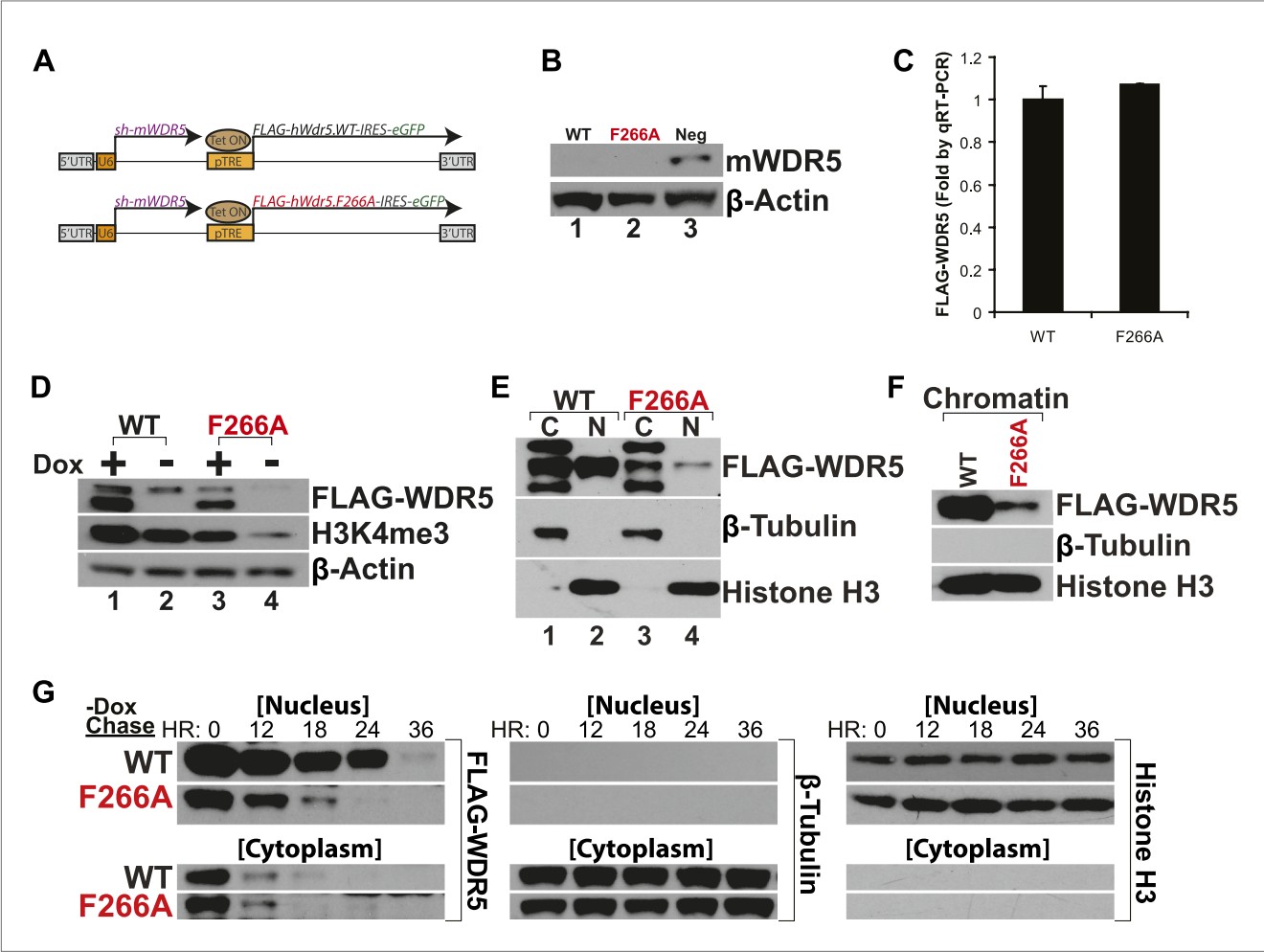

**Figure 3**. WDR5 F266A mutation decreases protein stability and localization to chromatin. (**A**) Schematic of lentiviral vectors, modified from (*Ang et al., 2011*). (**B**) Western blot demonstrating successful mouse WDR5 knockdown. (**C**) qRT-PCR results demonstrating equal RNA expression of human WDR5 WT and WDR5 F266A. (**D**) Western blot of WDR5 WT and WDR5 F266A protein expression, also with 4 days after doxycycline removal. (**E**) WDR5 F266A is defective in nuclear accumulation, compared with WDR5 WT. (**F**) WDR5 F266A reduces chromatin association, as seen in chromatin isolation experiments. (**G**) WDR5 F266A mutation decreases protein stability in the nucleus after doxycycline withdrawal. Both WDR5 WT and WDR5 F266A are similarly unstable in the cytoplasmic fraction.

As expected, both cell lines demonstrated similar knockdown of endogenous mouse WDR5 (*Figure 3B*), with equal transcription of exogenous human WDR5 RNA (*Figure 3C*). Unexpectedly, more human WDR5 WT protein was present when compared with human WDR5 F266A (*Figure 3D*). In WDR5 WT ESCs, promoter shutoff of WDR5 resulted in ~50% reduction of H3K4me3 (Lanes 1 and 2), as previously reported (*Ang et al., 2011*). Interestingly, ESCs harboring WDR5 F266A displayed a similar global 50% reduction of H3K4me3 (Lane 3), although WDR5 F266A protein was expressed at lower levels. In addition, promoter shutoff of WDR5 F266A caused a drastic >80% reduction of H3K4me3 levels (Lanes 3 and 4). Thus, WDR5 F266A exhibits a striking inability to maintain global H3K4me3 levels.

We more closely examined the subcellular localization of WDR5 WT and WDR5 F266A, and found that WDR5 F266A was depleted in both nuclear (*Figure 3E*) and chromatin fractions (*Figure 3F*). The alteration in cellular distribution of WDR5 F266A can be explained by differential stability in nuclear vs cytoplasmic pools, or alternatively, by a role of F266 in directly controlling nuclear localization. We believe that the former is the correct explanation based on the fact that fusion of F266A to a strong nuclear localization signal (GAL4 DNA binding domain) did not rescue its function (*Figure 2D*).

To further assess WDR5 protein stability directly, we conducted pulse chase experiments using doxycycline withdrawal to shutoff *FLAG-hWDR5* transcription, and followed the fate of pre-existing WDR5 WT or F266A protein in the nucleus and cytoplasm (*Figure 3G*). We found that in the cytoplasm, both WDR5 WT and F266A have equally short half-lives of less than 12 hr. In contrast, nuclear WDR5 WT persists for up to 24 hr, but WDR5 F266A is turned over at least 12 hr sooner (*Figure 3G*). Thus, ability to bind lncRNAs appears to be essential for WDR5 protein stability, suggesting a RNA-mediated post-translational regulation of WDR5.

## Over 1000 ESC RNAs bind WDR5

Our in vitro and in vivo data characterizing the WDR5 F266A mutant suggest that RNA binding is an important aspect of WDR5's cellular function, however currently only two RNAs have been identified as WDR5 partners. To close this gap, we identified WDR5–bound RNAs in the ESC transcriptome. We tested UV-crosslinking with PAR-CLIP, but found that WDR5-RNA interactions have poor inherent UV crosslinking ability. We also found that standard RNA immunoprecipitation (RIP) with FLAG epitope gave substantial background that hampered data interpretation. We then turned to RNA:protein immunoprecipitation in tandem (RIPiT), a method designed to identify RNA targets of RBP complexes with poor UV linking capacity (*Singh et al., 2013*). Specifically, we fused tandem FLAG and hemagglutinin (HA) tags to wild type WDR5 or the F266A mutant. We established ESC lines expressing FLAG-HA tagged version of WT and F266A WDR5 at near endogenous expression levels, and FPLC analysis confirmed that both tandem-tagged WDR5 proteins quantitatively formed equivalent MLL-WDR5 protein complexes with endogenous subunits (*Figure 4—figure supplement 1A,B*).

We performed RIPiT-seq with tandem immunopurified FLAG-HA-tagged WDR5 from formaldehyde crosslinked ESCs. To afford high stringency to the data, we searched for RNAs that were enriched in the WDR5 WT but not in the mock infected or WDR5 F266A datasets. WDR5 WT, but not F266A or control, is associated with 1434 RNAs, comprised of mRNAs, lncRNAs, pri-miRNAs, and snoRNAs (*Figure 4A,B*). Thus, a family of ESC RNAs bind WDR5 through the same surface as HOTTIP enhancer-like RNA that act in other cell types, dramatically expanding the known RNA targets of WDR5. WDR5 binds to 23 previously identified ESC lncRNAs (*Guttman et al., 2011*; *Ulitsky et al., 2011*; *Ng et al., 2012*; *Sheik Mohamed et al., 2010*). Importantly, six lncRNA partners of WDR5 have been shown to be required for ESC pluripotency or differentiation when individually depleted by shRNAs (*Figure 4B*, lncRNAs with ESC phenotype in orange) (*Guttman et al., 2011*). Prior protein interaction surveys with repressive chromatin modification complexes had failed to identify partners for these lncRNAs except *lincRNA-1592* (*Guttman et al., 2011*), and their interaction with WDR5-MLL provides an explanation for their functions in ESC biology.

Previous reports of RNA-WDR5 interaction suggest the possibility for both *cis*-acting tethering and *trans*-acting mechanisms (*Wang et al., 2011*; *Gomez et al., 2013*). To investigate whether and how these mechanisms are utilized genome-wide, we compared RNA interactors of WDR5 identified by RIPiT-seq vs chromatin occupancy sites of WDR5 identified by ChIP-seq. *Ang et al. (2011)* reported 9303 genes that are bound by WDR5 and other trithorax group proteins in ESCs with high confidence, and 590 of these loci produced RNAs that physically interacts with WDR5 (*Figure 4C*). Restricting the analysis to the top quintile of WDR5-bound promoters still showed very limited overlap (100 out

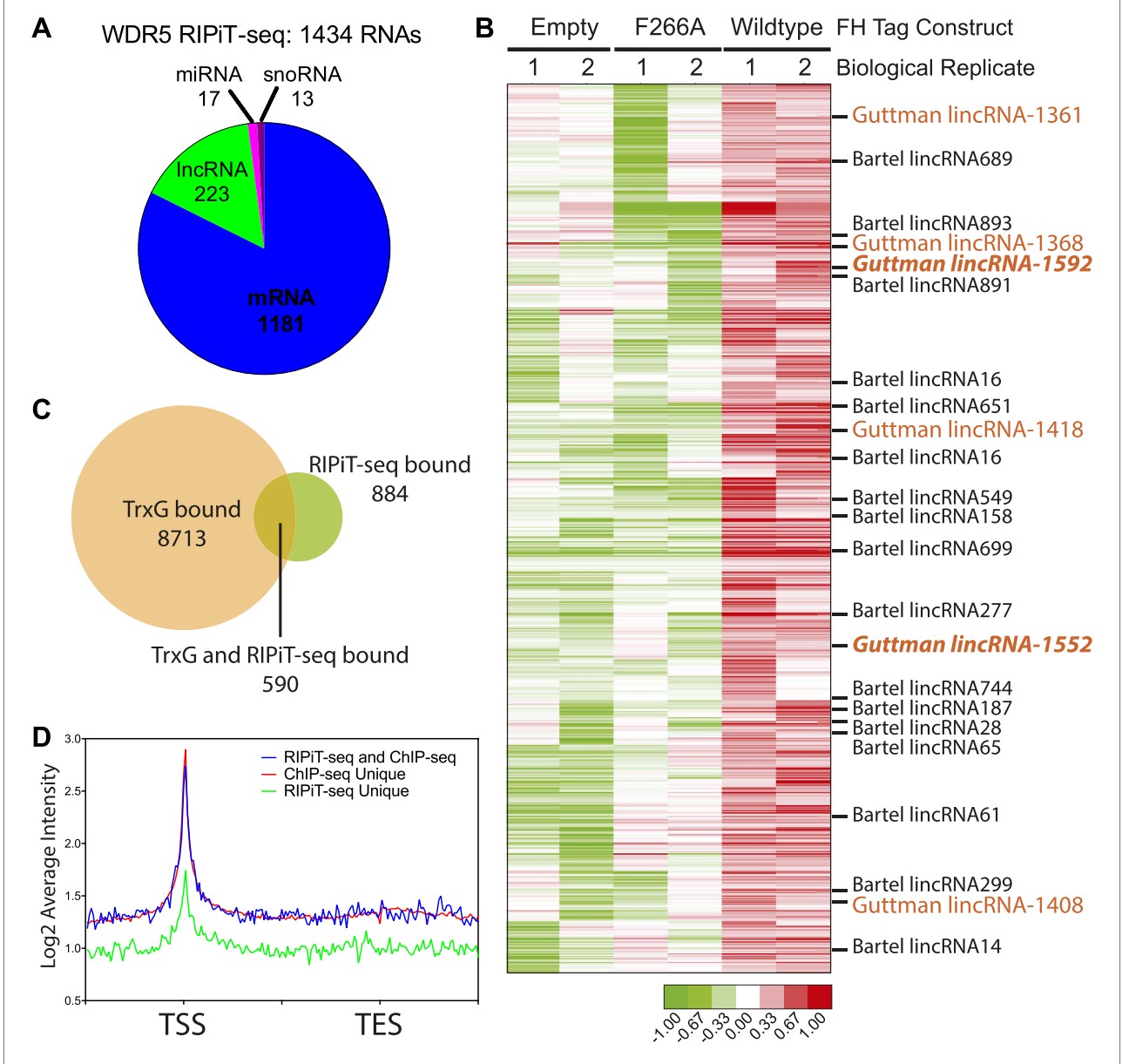

**Figure 4**. WDR5 binds to a family of ESC RNAs. (**A**) WDR5 RIPiT-seq retrieves diverse class of RNAs. (**B**) 1434 RNAs bind to wild type WDR5 but not F266A mutant or vector control. Each column is a RIPiT-seq experiment; each row is a transcript. Red indicates enrichment over input and WDR5 F266A mutant. Known ESC lncRNAs are listed; those with known functions in ESC pluripotency or differentiation are highlighted in orange. Two lincRNA loci that are both bound by WDR5 on chromatin and generate lincRNAs that bind WDR5 are highlighted in orange bold font. (**C**) Overlap of genes retrieved by WDR5 RIPiT-seq vs ChIPseq. (**D**) Metagene analysis of WDR5 ChIP-seq signal in the indicated classes of genes.

The following figure supplements are available for figure 4:

**Figure supplement 1**. Characterization of FLAG-HA ES cell lines.

of 1849 promoters produce RNAs that also bind WDR5). Plotting average WDR5 ChIP-seq signal at ChIP-seq only or genes with both ChIP-seq and RIPiT-seq signal revealed that their WDR5 binding profiles are indistinguishable, highlighting the value of the RIPiT data set to reveal RNA involvement (*Figure 4D*). In contrast, the class of genes with RIPiT signal only had substantially lower WDR5 chromatin occupancy at their promoters (*Figure 4D*); one or more of these RNAs may be decoys that remove WDR5 from cognate loci.

The RNAs that both bind to WDR5 and have strong WDR5 ChIP-seq signal at their promoters are candidates for additional *cis* regulators of WDR5 recruitment. Two such lncRNAs, *lincRNA-1552* and *lincRNA-1592*, were shown to previously impact ESC biology, and *lincRNA-1552* is a key node in the pluripotency gene regulatory network. Depletion of *lincRNA-1552* led to mis-expression of ~100 mRNAs including *Nanog* and *Pou5f1*, and the promoter of *linc-1552* is bound by Pou5f1, Klf4, Nanog, Zfx, n-Myc, and c-Myc (*Guttman et al., 2011*), suggesting a positive feedback loop has been described for other ESC lncRNAs (*Sheik Mohamed et al., 2010*). Further, the expression of *linc-1552* is repressed by retinoic acid-induced ESC differentiation (*Guttman et al., 2011*). This example highlights of how WDR5 can interact with a lincRNA important to ESC core transcription factor circuitry to control ESC pluripotency. Additionally, a previously reported screen in ESCs identified 901 protein-coding genes important for ESC pluripotency, and 23 of the corresponding mRNAs were enriched in WDR5 RIPiT-seq, including *Myc* and *Bmp4* mRNAs (*Ivanova et al., 2006*) (*Supplementary file 1*). Our results also imply many ESC promoters can recruit WDR5 by mechanisms independent of RNA, consistent with other reports (*Ang et al., 2011*; *Jiang et al., 2011*). The discovery of many RNAs that bind WDR5 but not at its cognate genomic loci suggests that some WDR5-binding RNAs do not bind WDR5 as nascent transcripts, but only interact with WDR5 at a later stage of maturation and possibly for action in *trans*. Thus, WDR5 binds over 1400 endogenous RNAs in ESCs with potential for both *cis* and *trans* chromatin regulation.

## RNA binding to WDR5 is essential for maintenance of embryonic stem cell fate

Loss of RNA binding also impacts the ability of WDR5 to promote ESC self-renewal. (*Figure 5A,B*). In WDR5 WT ESCs, promoter shut-off caused a ~50% reduction in the number of alkaline phosphatase-positive colonies, a specific indicator of ESC state. Notably, F266A WDR5 ESCs grown in the presence of doxycycline displayed a similar ~50% reduction in colony number. By morphology, both WDR5 WT ESCs–dox and WDR5 F266A ESCs +dox formed similar small, partially differentiated colonies. Doxycycline withdrawal in WDR5 F266A ESCs caused a >90% reduction in number of colonies, with the majority of cells forming clusters of fully differentiated alkaline phosphatase-negative cells.

WDR5 directly binds to the chromatin of key genes required for in ESC self renewal (*Ang et al., 2011*). Genome-wide expression profiling further confirmed that WDR5 F266A was unable to sustain the gene expression program associated with ESC self-renewal and pluripotency, and instead allowed ectopic expression of genes indicative of ectodermal and mesodermal lineages (*Figure 5C*). Validation of microarray results by qRT-PCR further demonstrated that ESCs expressing WDR5 F266A had significantly reduced mRNA levels of pluripotency regulators *Pou5f1*, *Nanog*, *Sox2*, and *Esrrb*, but upregulation of differentiation markers *Cdx2*, *Fgf5*, and *Nestin* (*Figure 5D,E*).

## Discussion

### Identification of a WDR5-RNA interface

Guided by the crystal structure and comprehensive mutagenesis, we identified the lncRNA binding cleft in WDR5. WDR5 is a multifunctional adaptor protein that can interact with multiple subunits of the MLL complex, such as MLL itself and RbBP5, as well as with regulators and substrates of the complex, such as HOTTIP and histone H3 arginine 2 symmetric demethylation (H3R2me2s) (*Migliori et al., 2012*). The MLL and H3R2me2s share the same binding pocket, and we found that HOTTIP and RbBP5 share a distinct binding pocket. While Pou5f1 has been reported to bind WDR5, we found no difference in the WDR5 F266A lncRNA binding mutant for its interaction with Pou5f1 (data not shown). Nonetheless, these multitudes of interactions suggest that molecular arrangements in this complex may be dynamic and intricate.

By strategic structure-guided mutagenesis, we identified WDR5 F266A as a mutant that has a specific defect in binding the lncRNA HOTTIP, but does not impact interactions with MLL1, RbBP5, or Ash2L. More importantly, the F266A mutation does not diminish MLL1 complex catalytic activity in vitro. Thus, although HOTTIP and RbBP5 share an overlapping binding pocket, the specific residues required for binding appear fundamentally different. This result is not surprising, given the electrostatic and three dimensional structure differences between peptides vs RNAs. The WDR5 F266A mutant is largely deficient in gene activation in cells, suggesting that direct lncRNA-protein interactions are important for WDR5 function in vivo.

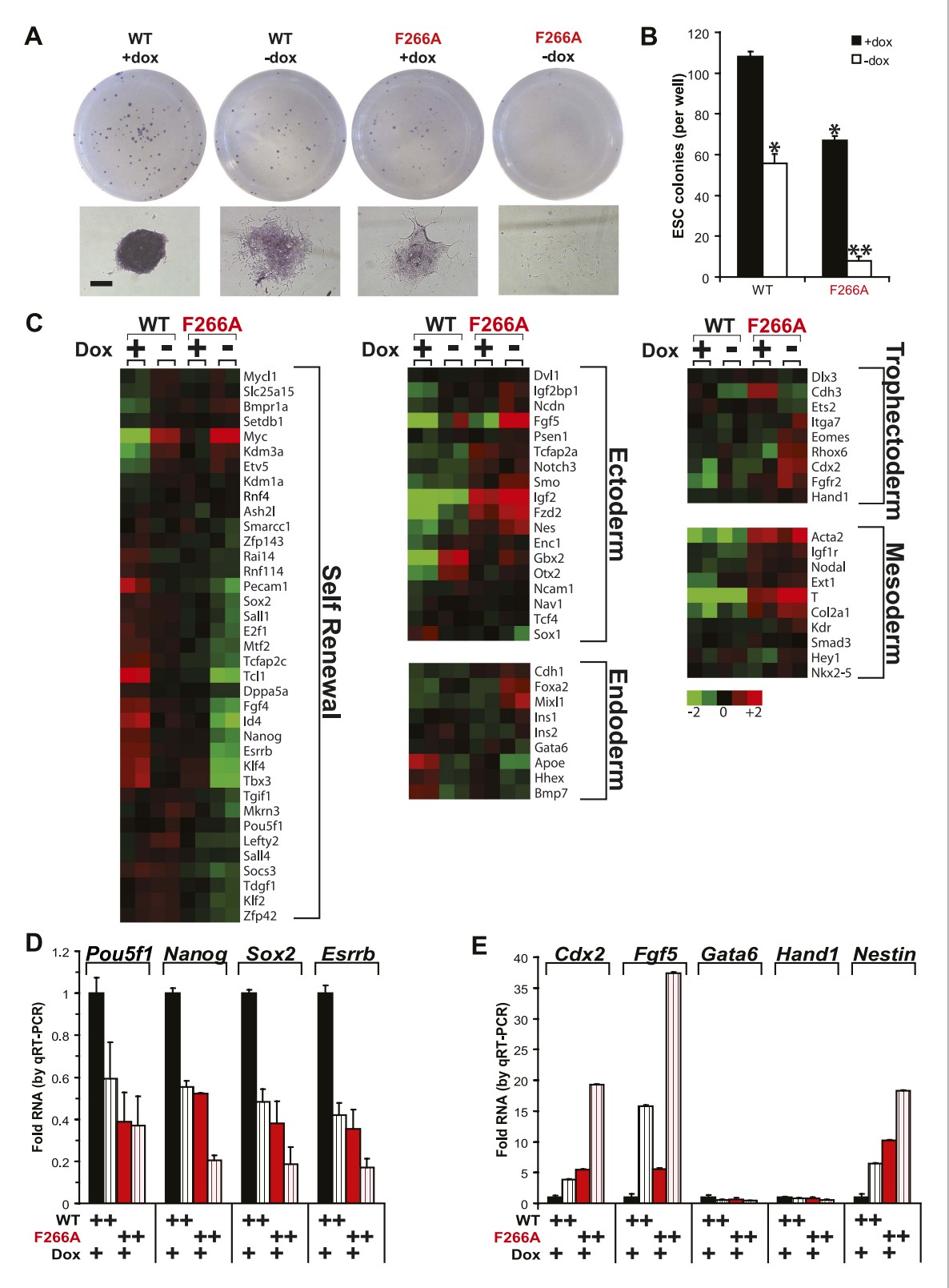

**Figure 5**. WDR5 F266A mutation causes defects in ESC self renewal and increases differentiation. (**A** and **B**) Alkaline phosphatase staining and morphology of ESC colonies after 6 days growth in conditions specified. WDR5 F266A demonstrates reduced alkaline phosphatase positive colonies with increased differentiation compared with wild type. (**A**) Representative wells and colony morphology. Scale bar represents 200 μm. (**B**) Quantitation of
*Figure 5. Continued on next page*

*Figure 5. Continued*

colonies per well. (**C**) Microarray data reveal that both WDR5 F266A +dox and WDR5 WT −dox displayed loss of self renewal genes. WDR5 F266A +dox further demonstrates increased expression of ectodermal and mesodermal markers. Mixl1 is a mesendoderm marker indicative of potential for both endoderm and mesoderm. (**D** and **E**) qRT-PCR validation of (**D**) reduced expression of ESC pluripotency genes and (**E**) ectopic expression of differentiation genes with either WDR5 F266A expression or shutoff of WDR5 WT. qRT-PCR values were first normalized to the housekeeping gene beta-actin, then normalized to WT. Mean +/− SD are shown.

## Requirement of a RNA-binding interface in WDR5 for gene activation in ESC self renewal

We replaced endogenous WDR5 in mouse ESCs with a RNA binding mutant, WDR5 F266A, and found that the RNA binding interface is critical for the accumulation of nuclear and chromatin WDR5. These results suggest that in addition to their roles in recruiting protein complexes to specific genomic loci, lncRNA binding may also regulate the longevity of their interacting protein complexes (*Figure 6*). When able to bind RNAs, wild type WDR5 displays a long nuclear and chromatin half-life, thus allowing formation of the MLL1 complex for H3K4me3 and gene activation of ESC pluripotency genes. However, when unable to bind RNAs, WDR5 mutant is rapidly turned over, causing loss of H3K4me3 and reduced expression of pluripotency genes. Thus, RNA binding may directly or indirectly regulate protein turnover for epigenetic regulation. It is possible that WDR5 protein that is not properly complexed or localized on chromatin are degraded by default, or WDR5 degradation may be a regulated mechanism (*Nakagawa and Xiong, 2011*). In our study, ability to bind RNAs extends WDR5 protein half-life by an additional 12 hr to approximately 24 hr, which starts to approach the time scale of an entire cell division cycle (especially in ESCs). With the extended protein longevity, lncRNAs may then regulate protein occupancy on chromatin during cellular maintenance, as well as aid in transmission of epigenetic regulators across cell generations. In this manner, lncRNAs may play crucial roles in regulating the multigenerational transmission of epigenetic proteins required for maintenance of cell state.

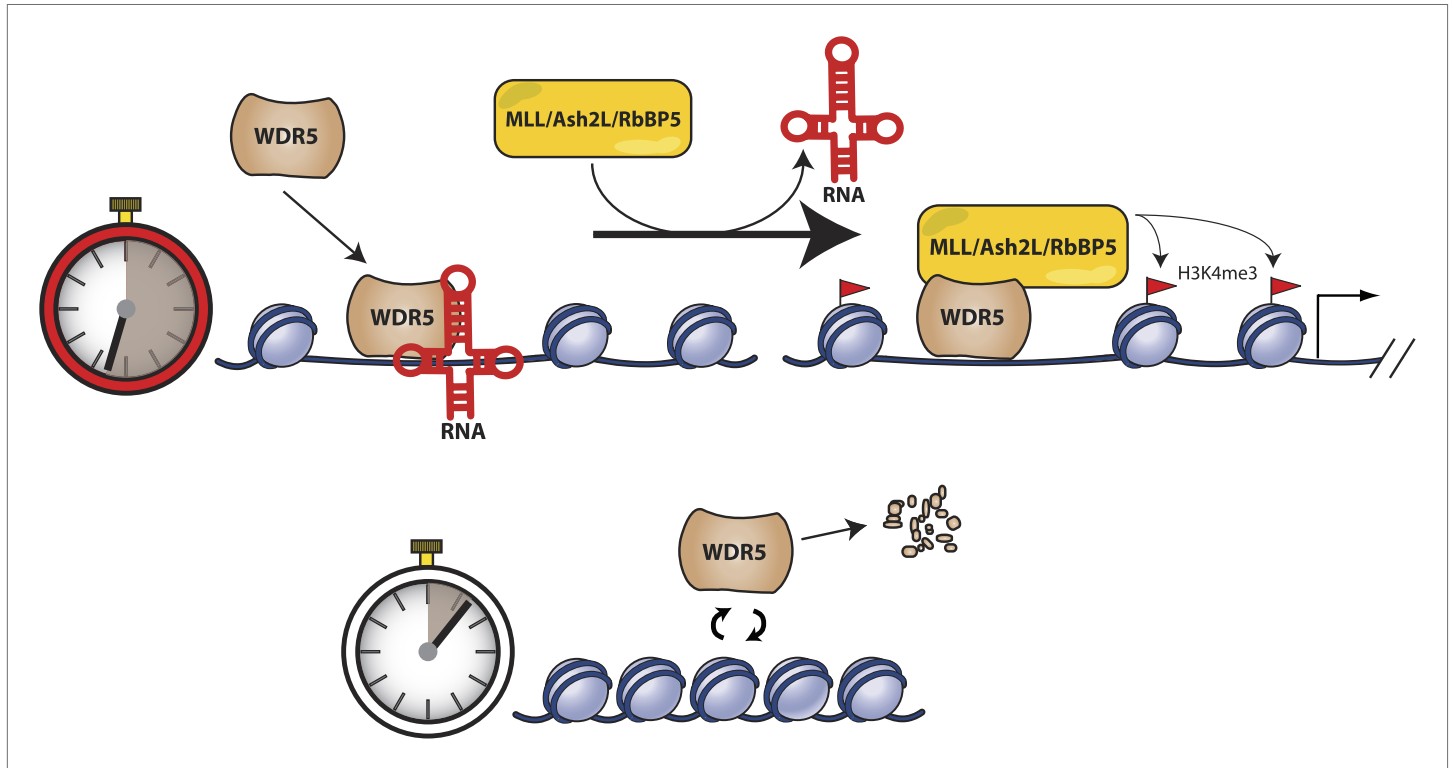

**Figure 6**. Model for RNA-mediated switch of protein turnover to activate target genes. Expressed lncRNAs bind the WDR5 protein to increase protein stabilization on chromatin, which facilitates MLL complex assembly and methyltransferase activity for target gene activation. Without lncRNA binding, WDR5 fails to associate effectively with chromatin and is rapidly degraded.

Given the dynamic regulation of nucleosome/chromatin mark turnover, protein stability also greatly affects the epigenetic landscape within the cell. Nucleosomes are turned over in a dynamic and regulated fashion at specific genomic loci (*Dion et al., 2007*; *Mito et al., 2007*; *Deal et al., 2010*), and histone tails can be proteolyzed during differentiation to erase prior histone modifications (*Duncan et al., 2008*). In our example, the loss of stabilized WDR5 leads to an inability to maintain global H3K4me3 level, loss of expression of pluripotency regulators, and loss of ESC self renewal. Interestingly, the impact of F266A mutation on H3K4me3 level is modest in steady state conditions, but becomes quite striking with transcriptional shutoff of WDR5. We interpret these results to indicate that WDR5 stability is important for the robustness of the active chromatin state, exemplified by H3K4me3. In the absence of stabilized WDR5, H3K4me3 becomes much more sensitive to fluctuations in the transcriptional levels of WDR5 and likely other subunits of the MLL complex. Thus, these results reveal a new facet for RNA regulation in the connection between protein half-lives and the temporal transmission of epigenetic information via chromatin.

## A family of RNA partners for WDR5-MLL complex

RIPiT-seq reveals nearly 1500 cellular RNAs in ESCs that are associated with WDR5 complex in a manner dependent on the WDR5 RNA binding pocket. These results support and substantially broaden the scope of RNAs that may impact WDR5's function, including many lncRNAs and mRNAs that collectively may equal the copy number of WDR5 protein. The WDR5 F266A lncRNA-binding deficient mutant provides an ideal reagent to act as nonbinding control in the systematic genome-wide identification of functional WDR5-binding RNAs. Our data find that some of the recent discovered lncRNAs important for ESC self-renewal and pluripotency are associated with WDR5 (*Guttman et al., 2011*; *Chakraborty et al., 2012*), including five lncRNAs with ESC phenotypes that do not bind repressive chromatin complexes. However, there are numerous additional RNAs that have not been characterized with respect to cell fate control or chromatin function, and thus suggest additional layers of regulation at the level of lncRNAs. Comparative analysis of genomic locations of WDR5 occupancy and WDR5 RNA binding suggest configurations that are compatible with both *cis* or *trans* regulation. Studies of other chromatin modification complexes have shown that a single complex (e.g., PRC2) can interact with thousands of RNAs, each of which can regulate a subset of genes targeted by the complex (*Khalil et al., 2009*; *Zhao et al., 2010*). Additionally, promiscuous binding of RNAs may allow PRC2 and other silencing factors to survey the genome and distinguish transcribed vs non-transcribed regions (*Davidovich et al., 2013*; *Di Ruscio et al., 2013*; *Kaneko et al., 2013*). A similar model likely applies to MLL and other gene activating complexes as well. The MLL complex possesses multiple components with RNA-binding capabilities. While WDR5 binds HOTTIP, NeST, and many other RNAs as shown here (*Wang et al., 2011*; *Gomez et al., 2013*), MLL1 itself also has domains that bind RNAs in specific and nonspecific fashions (*Krajewski et al., 2005*; *Bertani et al., 2011*). Cyp33, an allosteric regulator of MLL, also contains a RNA recognition motif, and its regulation of MLL can be controlled in an RNA-dependent manner (*Hom et al., 2010*; *Wang et al., 2010*). The presence of multiple RNA interaction domains suggests intimate and potentially multiple roles for RNAs in the control of H3K4me3 and programming active chromatin.

## Materials and methods

### Cell lines, antibodies, and vectors

HEK293T/17 cells were obtained from the American Type Culture Collection (ATCC). The HEK293T 5XUAS-Luciferase cell line has been previously described (*Wysocka et al., 2005*). V6.5 mouse embryonic stem cells were grown on 0.2% gelatinized plates in Knockout DMEM supplemented with 15% FBS, 1% Glutamax (35050; Invitrogen, Carlsbad, CA), 1% nonessential amino acids, 1% Pen/Strep, 0.2% β-mercaptoethanol, LIF (ESGRO1107; 1:10000; Millipore, Billerica, MA), and 2 µg/ml doxycycline (D9891; Sigma, St. Louis, MO). To maintain consistent levels of doxycycline, media was changed every 2 days. Antibodies used were: anti-FLAG-M2 (F1804; Sigma, St. Louis, MO), anti-β actin (ab8227; Abcam, Cambridge, MA), anti-RbBP5 (A300-109A; Bethyl, Montgomery, TX), anti-H3K4me3 (ab8580; Abcam, Cambridge, MA), anti-β tubulin (ab6046; Abcam, Cambridge, MA), anti-H3 (ab1791; Abcam, Cambridge, MA), anti-WDR5 (07-706; Millipore, Billerica, MA), and anti-HA (MMS-101P; Covance, Princeton, NJ). All immunoprecipitations were conducted using anti-FLAG-M2 (A2220; Sigma, St. Louis, MO) or mouse IgG (A0919; Sigma, St. Louis, MO) agarose beads. Unless noted, all expression vectors were cloned into pcDNA3 or pcDNA3.1 + backbones.

## qRT-PCR primers:

Hottip (F:CCTAAAGCCACGCTTCTTTG, R:TGCAGGCTGGAGATCCTACT)
Hist1h2bg (F:GGCATCATGAATTCCTTCGT, R:GCTTGTTGTAGTGGGCCAGA)
Hotair (F:GGTAGAAAAAGCAACCACGAAGC, R:ACATAAACCTCTGTCTGTGAGTGCC)
Flag-Wdr5 (F:GACTACAAAGACGATGACGACAA, R:TCCCAGCTTGTGACCAGATA)
Gapdh (F:AGGTGGAGGAGTGGGTGTCGCTGTT, R:CCGGGAAACTGTGGCGTGATGG)
Rnu1 (F:ATACTTACCTGGCAGGGGAG, R:CAGGGGGAAAGCGCGAACGCA)
Actb (F:GCTGTATTCCCCTCCATCGTG, R:CACGGTTGGCCTTAGGGTTCAG)
Esrrb (F:GGGTAGAGCCCACTTGTTCA, R:AGGTAGCCTGGGTTTTTGCT)
Cdx2 (F:CCTGCGACAAGGGCTTGTTTAG, R:TCCCGACTTCCCTTCACCATAC)
Fgf5 (F:AAAGTCAATGGCTCCCACGAA,R:GGCACTTGCATGGAGTTTTCC)
Gata6 (F:ACAGCCCACTTCTGTGTTCCC, R:GTGGGTTGGTCACGTGGTACAG)
Hand1 (F:GCGTCAGTACCCTGATGCCTTC, R:AAAGAGGAGGTAAGAGGACGGAAG)
Nanog (F:TGGTCCCCACAGTTTGCCTAGTTC, R:CAGGTCTTCAGAGGAAGGGCGA)
Nestin (F: AGGCGCTGGAACAGAGATT, R: TTCCAGGATCTGAGCGATCT)
Pou5f1 (F:GTGGAGGAAGCCGACAACAATGA, R:CAAGCTGATTGGCGATGTGAG)
Sox2 (F:CAGGAGAACCCCAAGATGCACAA, R:AATCCGGGTGCTCCTTCATGTG)

## In vitro GST pulldown and competition

GST pulldown and competition assays were performed as previously described, with modifications (*Wang et al., 2011*). GST-tagged WDR5 wild type, mutants, and GST control proteins were expressed and purified from *E. coli* as previously described (*Smith and Johnson, 1988*). GST proteins were bound to glutathione sepharose 4B (Amersham/GE Healthcare, Pittsburgh, PA), blocked with 0.2 mg/ml BSA (Ambion, Carlsbad, CA) in 2X binding buffer (40 mM HEPES pH 7.6, 200 mM KCl) at 4°C for 1 hr. For peptide competition assays, indicated peptides (Elim Biopharmaceuticals) were also added. 0.04 mg/ml heparin (H3149; 1:1; Sigma, St. Louis, MO) was added to further block nonspecific RNA binding. Protein-bound beads were then incubated with folded T7-transcribed HOTTIP or histone 1H2BG mRNA for 45 min (T7; Promega, Madison, WI). Beads were washed twice with PB200 (20 mM HEPES pH 7.6, 200 mM KCl, 0.05% NP-40) and once with DEPC-treated water to remove detergents. After resuspension in water, beads were directly used in qRT-PCR reactions to determine RNA levels. To confirm lack of degradation, bead-bound proteins were analyzed by Silver Stain Plus (Bio-Rad, Hercules, CA).

## Native RNA immunoprecipitation (RIP)

Native cell-based RNA immunoprecipitation was performed as previously described with modifications (*Dignam et al., 1983*). 293T cells were transfected with pcDNA3.1+FLAG-WDR5 and RNA expression plasmid. After 48–72 hr, cells were harvested by scraping in cold PBS, spun down, and pellets were then snap frozen in liquid nitrogen and stored at −80°C. Cells were resuspended in Buffer A (10 mM HEPES pH 7.5, 1.5 mM $MgCl_2$, 10 mM KCl, 0.5 mM DTT, 1.0 mM PMSF, supplemented with RNaseOUT [1:100; Invitrogen, Carlsbad, CA]), and cell membranes were lysed with 0.25% NP-40. After centrifugation, resulting nuclei were further lysed in Buffer C (20 mM HEPES pH 7.5, 10% glycerol, 0.42M KCl, 4 mM $MgCl_2$, 0.5 mM DTT, 1.0 mM PMSF supplemented with RNaseOUT [1:100]). Resulting whole cell lysates were immunoprecipitatied with mouse anti-FLAG-M2 or control mouse IgG agarose beads, and washed four times with wash buffer (50 mM Tris–HCl pH 7.9, 10% glycerol, 100 mM KCl, 5 mM $MgCl_2$, 10 mM β-mercaptoethanol, 0.1% NP-40). To confirm immunoprecipitation, FLAG peptide-eluted protein-RNA complexes were analyzed by western blot. Coimmunoprecipitated RNA was extracted using Trizol LS (Life Technologies, Carlsbad, CA) and Qiagen RNeasy columns, treated with TURBO DNAfree, and then analyzed by qRT-PCR (Brilliant II Sybr Green). Resulting values were defined as fraction of input, then normalized to the positive control sample.

## Formaldehyde crosslinked RIPiT-seq

RIPiT-seq was performed large as described previously (*Singh et al., 2013*) with the following modifications. ESCs were fixed for 15 min with 1% formaldehyde in 1xPBS at 25°C. The reaction was stopped by added 1/10th the volume of 1.25M Glycine. For a typical RIPiT-seq experiment chromatin was

prepared from ~1 × 10⁷ cells by thawing previously crosslinked ESC pellets and resuspending in 880 µl of Nuclear Lysis Buffer (50 mM Tris–HCl pH 7, 10 mM EDTA, 1% SDS) supplemented with fresh Protease Inhibitor Cocktail (Roche, Indianapolis, IN), 10 mM PMSF, and RNaseOUT (1:100). Chromatin was sheared in a Covaris E220 ultrasonicator to a size of 200–500 bp. The first IP was performed with 50 µl of FLAG-M2 agarose slurry (Sigma, St. Louis, MO) for 4 hr at 4°C on rotation. Samples were washed one time 1 ml in series at 4°C for 5 min each with the following buffers: High Stringency Buffer (15 mM Tris–HCl pH 7, 5 mM EDTA pH 8, 2.5 mM EGTA pH 8, 1% Triton-X100, 1% Sodium Deoxycholate, 120 mM NaCl, 25 mM KCl), High Salt Buffer (15 mM Tris–HCl pH 7, 5 mM EDTA pH 8, 2.5 mM EGTA pH 8, 1% Triton-X100, 1% Sodium Deoxycholate, 1M NaCl) and NT2 Buffer (50 mM Tris–HCl pH 7, 150 mM NaCl, 1 mM MgCl₂, 0.0005% NP-40). Captured FH-WDR5 complexes were then eluted from the FLAG-M2 agarose by incubating each sample in 500 µl of FLAG Elution Buffer (50 mM Tris–HCl pH 7.5, 250 mM NaCl, 0.5% NP-40, 0.1% Sodium Deoxycholate) and 0.5 µg/ml FLAG Peptide (PAN Facility, Stanford University) for 25 min at 4°C on rotation. The FLAG elution was repeated once, achieving a total elution volume of 1 ml. FH-WDR5 complexes were then captured by adding 10 µl HA-agarose slurry (Pierce) to each sample for 1 hr at 4°C on rotation. The samples were then washed as previously done on the FLAG-M2 beads. After washing each sample was resuspended in 100 µl of ProteinaseK Buffer (10 mM Tris–HCl pH 7, 100 mM NaCl, 1 mM EDTA pH 8, 0.5% SDS) and 5 µl of ProteinaseK (Ambion, Carlsbad, CA) for 45 min at 50°C in a Thermomixer (Eppendorf, Hauppauge, NY) at 1000 rpm. After ProteinaseK treatment 500 µl of Trizol LS (Qiagen, Venlo, Limburg) was added to each sample, vortexed for 10 s, and incubated at 25°C for 5 min. Total RNA was recovered from each sample with the miRNeasy kit (Qiagen, Venlo, Limburg) following the manufacturer's instructions, treated with TURBO-DNAfree (Ambion, Carlsbad, CA), and repurified with the miRNeasy kit. The Ovation v2 Kit (NuGEN, San Carlos, CA) was used according to the manufactures instructions to produce dsDNA from the purified RNA enriched by the RIPiT. DNA isolated from the Ovation v2 Kit was then used in the NEBNext ChIP-Seq Library Prep Master Mix Set for Illumina Kit (NEB, Ipswich, MA) following the manufactures instructions to produce barcoded deep sequencing libraries. Library PCR material was quality checked and quantified on an Agilent BioAnalyzer 2100 and samples were sequenced with a run type of 1 × 50 bp on an Illumina HiSeq 2500 machine.

### RIPiT-seq data processing and analysis

Sequencing reads were mapped to the mouse genome (mm9 assembly) using TopHat (version 1.1.3) (*Trapnell et al., 2009*). Each sample generated 13.6 to 51.1 million mapped sequences, recorded in BAM/SAM format. A non-redundant mm9 transcriptome was assembled from UCSC RefSeq genes, UCSC genes, and predictions from (*Ulitsky et al., 2011*) and (*Guttman et al., 2011*). Gene expression in the form of log2d RPKM was calculated using a self-developed script. An enrichment score of each gene in each sample was defined as the fold change of log2d RPKM between the pull down sample vs its corresponding input. A gene was defined enriched in wide-type WDR5 pull down if the minimum enrichment score of this gene in both wide-type replicates is greater than 0 and greater than the maximum of that in empty vector and mutated samples. There were 1434 enriched genes, whose enriched score were clustered using Cluster and plotted in Treeview. WDR5 ChIPseq peaks were downloaded from *Ang et al. (2011)*, and genes whose promoter region (2 kb upstream and 1 kb downstream of its TSS) overlaps with any WDR5 ChIPseq peak were defined as WDR5 bounded genes. Overlay of WDR5 bounded genes with WDR5 RIPseq enriched genes was shown and p value of the significance was estimated using Fisher's Exact test. RIPiT data are available at Gene Expression Omnibus under the accession number GSE53035.

### Isothermal titration calorimetry

The *Kd* values of the WDR5-RbBP5$_{330–381}$, WDR5-MLL1$_{3762–3802}$, WDR5-H3$_{1–10}$ were determined by a ITC200 calorimeter (MicroCal) in buffer 25 mM Tris, 150 mM NaCl, pH 8.0. The latter components were used as titrants. The concentration of macromolecule solution is 60–100 µM and the concentration of titrant is 600–1000 µM. ITC data were analyzed and fit using Origin 7 (OriginLab).

### GST protein pull-down assays

40 µg GST-fused WDR5 (WT or mutants) mixed with MLL1$_{SET}$, Ash2L, RbBP5 were incubated with 10 µl glutathione-Sepharose 4B beads for 2 hr at 4°C. After extensive wash, the bound protein were eluted in 50 mM Tris–HCl, pH 8.0, 300 mM NaCl, 15 mM reduced glutathione. The input samples and eluted samples were loaded on 12% SDS-PAGE and Coomassie stained.

## Histone methylase activity assay and SAM binding assay

Reactions were carried out at room temperature for 1 hr in the presence of [$^3$H]-SAM (S-adenosyl-L-[methyl-$^3$H] methionine) as previously described (*Cao et al., 2010*). 0.25 mM unmodified H3K4 peptides (10 mer:ARTKQTARKS) was used as substrate. 0.5 µM of WDR5, RbBP5, Ash2L and MLL1$_{SET}$ proteins were used. Each assay was performed in triplicate.

## Luciferase assay

pCMX-GAL4-WDR5 and pRL Renilla Luciferase plasmid were transfected into HEK293T 5XUAS-Luciferase cells using Lipofectamine 2000. 40–48 hr after transfection, cells were harvested and analyzed using the Dual-Luciferase Reporter Assay System (Promega, Madison, WI). Luciferase readings were normalized to Renilla readings.

## Establishment of FLAG-tagged WDR5 rescue ESCs

Lentiviral constructs were generated that that constitutively knock down mouse WDR5 while allowing expression of a doxycycline-inducible human WDR5. The rescue vector pLKO.tre (*Ang et al., 2011*) was modified by replacing the shRNA-immune mouse WDR5-cDNA with either wild type or F266A mutated human WDR5. The resulting pLKO vectors and rtTA (from Marius Wernig, Stanford University) constructs were cotransfected with packaging plasmids (pLKO: second generation pCMV-dR8.2 dvpr and pCMV-VSVG| rTta: third generation pMDLg/pRRE, pCMV-VSVG, pRSV-Rev) into 293T cells using Lipofectamine 2000 (Invitrogen, Carlsbad, CA). After 16 hr, media was changed. Viruses were harvested after 48 and 72 hr, pooled, and then clarified by centrifugation. Viruses were concentrated using Lenti-X concentrator, incubated with 8 µg/ml polybrene (Sigma H9268, St. Louis, MO) and then used to infect V6.5 ESCs. GFP+ cells were sorted (Stanford Shared FACS Facility) and then cultured as described above. For RNA analysis, 5000 GFP+ cells were plated in media with or without doxycycline, and then harvested after 6 days. For qRT-PCR analysis, resulting values were normalized to control β actin levels, and then normalized to WT+dox.

## Establishment of FLAG-HA-tagged WDR5 wild type and F266A ESCs

Lentiviral constructs were generated that contained either no cDNA, N-terminally FLAG-HA tagged WT WDR5 (human), or N-terminally FLAG-HA tagged WDR5 F266A (human) mutant. The F266A mutant construct has a 20 amino acid N-terminal truncation, which does not affect in vivo function or MLL complex formation. The parent vector was N103 (kind gift from Dr Jerry Crabtree Lab) and contains a puromycin resistance cassette. The resulting modified N103 vectors were reverse transfected with packaging plasmids MD2G and PSPAX2 (kind gift from Dr Jerry Crabtree Lab) into 293T cells using FuGENE HD (Promega, Madison, WI) according to the manufactures protocol. After 16 hr, media was changed. Viruses were harvested after 48 and 72 hr, pooled, and then clarified by centrifugation. Viruses were concentrated using Lenti-X concentrator, incubated with 8 µg/ml polybrene (H9268; Sigma, St. Louis, MO) and then used to infect V6.5 ESCs. Two days after infection clones with successful integration were selected by added 1 µg/ml Puromycin to ESC media. ESC colonies were screened for expression of the FH-WDR5 constructs and only clones with near endogenous levels of the FH-WDR5 or FH-WDR5-F266A proteins were selected.

## Fast performance liquid chromatography analysis of FH-WDR5 ESCs

After isolation and expansion of clonal populations of ESCs with near endogenous levels of FH-tagged wild type or F266A mutant WDR5 nuclei were isolated as described above. Nuclei were lysed in 300 µl of FPLC Buffer (50 mM Tris–HCl pH 7.5, 250 mM NaCl, 10% glycerol, and 0.1% NP-40). The lysates were loaded onto a HiPrep Sephacryl S-500 HR Gel Filtration Column (GE Healthcare Life Sciences, Pittsburgh, PA) using a Pharmacia Biotech FPLC pump system. Fractions (330 µl per fraction) were collected on ice and then subjected to SDS-PAGE and western blot analysis.

## Embryonic stem cell fractionation

ESCs were fractionated as previously described (*Tee et al., 2010*), with modifications. To isolate nuclei and cytoplasm, Pellets were lysed in nuclei isolation buffer (NIB: 10 mM Tris pH 7.5, 60 mM KCl, 15 mM NaCl, 1.5 mM MgCl$_2$, 1 mM CaCl$_2$, 250 mM Sucrose, 10% glycerol, and 0.1% NP-40). Lysates were centrifuged at low speed to separate cytoplasm (supernatant) from nuclei (pellet). Cells were washed once with NIB without NP-40, resuspended in NIB without NP-40, and treated with Turbo DNase

(Ambion, Carlsbad, CA). EDTA was added to a final concentration of 10 mM to neutralize the DNase. After centrifugation, the resulting washed pellet was resuspended in NIB, and NaCl was added to a final concentration of 500 mM to lyse the nuclei.

For chromatin isolation, cells were lysed with cytoskeleton buffer (CSK: 10 mM PIPES KOH pH 7, 100 mM NaCl, 300 mM sucrose, 3 mM MgCl$_2$, 0.5% Triton X-100), centrifuged, and the resulting pellet was washed twice more with CSK buffer. The washed pellet was resuspended in CSK buffer, and treated with Turbo DNase to release chromatin-bound proteins.

## Alkaline phosphatase staining

For colony forming assays, FLAG-tagged WDR5 rescue v6.5 ESCs were counted by flow cytometer, as well as hemocytometer, and 250 cells were plated onto 12 well plates with full media with or without doxycycline. After 6 days, cells were fixed with 1:1 methanol:acetone, then stained using Vector Blue (Vector Laboratories, Burlingame, CA) per the manufacturer's instructions.

## Microarray analysis

cDNA was synthesized, labeled, and hybridized to Affymetrix Mouse 430 2.0 arrays in biologic duplicates (Stanford Protein and Nucleic Acid Facility). Arrays were normalized by robust multi-array average (RMA) using justRMA package in R, and probes that had an expression value greater than 50 in at least one sample were defined as expressed probes. For each expressed probe, its expressions were log2ed, and the gene expression was defined as the average expression of all the expressed probes that attached to this gene. Differential expression was performed using significance analysis of microarrays (SAM) 3.0 (*Tusher et al., 2001*) with a false discovery rate less than 5%, an average fold change ≥2 in any compared groups. Heatmaps show mean-centered gene expression, based on previously described expression patterns (*Ang et al., 2011*). Microarray data are available at Gene Expression Omnibus under the accession number GSE36513.

## Acknowledgements

We thank members of Chang, M Wernig, and M Kay labs for assistance. We also thank IR Lemischka (Mount Sinai School of Medicine) for mouse shRNA tet constructs, J Wysocka (Stanford) for the HEK293T 5XUAS-Luciferase cell line, and GR Crabtree for lentiviral vectors. We also thank Ngon Nguyen and Peter Marinkovich for assistance with FPLC analysis. Supported by Medical Scientist Training Program (YWY, RAF), NIH (HYC), and California Institute for Regenerative Medicine (HYC). ML and HYC are Early Career Scientists of the Howard Hughes Medical Institute.

## Additional information

### Funding

| Funder | Author |
| --- | --- |
| Howard Hughes Medical Institute | Yul W Yang, Ryan A Flynn, Yong Chen, Kun Qu, Bingbing Wan, Kevin C Wang, Ming Lei, Howard Y Chang |
| California Institute for Regenerative Medicine | Yul W Yang, Ryan A Flynn, Kun Qu, Kevin C Wang, Howard Y Chang |
| National Institutes of Health | Howard Y Chang |

The funders had no role in study design, data collection and interpretation, or the decision to submit the work for publication.

### Author contributions

YWY, RAF, Conception and design, Acquisition of data, Analysis and interpretation of data, Drafting or revising the article; YC, BW, Acquisition of data, Analysis and interpretation of data; KQ, Analysis and interpretation of data, Drafting or revising the article; KCW, Conception and design, Analysis and interpretation of data; ML, HYC, Conception and design, Analysis and interpretation of data, Drafting or revising the article

## Additional files

### Supplementary files

• Supplementary file 1. Protein coding genes common between RIPiT-seq Enriched RNAs and mRNAs identified from *Ivanova et al. (2006)*.

### Major dataset

The following previously published dataset was used:

| Author(s) | Year | Dataset title | Dataset ID and/or URL | Database, license, and accessibility information |
|---|---|---|---|---|
| Ang Y, Tsai S, Lee D, Monk J, Su J, Ratnakumar K, Darr H, Chang B, Rendl M, Schaniel C, Bernstein E, Lemischka IR | 2011 | Wdr5 mediates self-renewal and reprogramming via the embryonic stem cell core transcriptional network | GSE19588; http://www.ncbi.nlm.nih.gov/geo/query/acc.cgi?acc=GSE19588 | Publicly available at the Gene Expression Omnibus (http://www.ncbi.nlm.nih.gov/geo/). |

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
