## [Decision Letter]

[Editors’ note: a previous version of this study was rejected after peer review, but the authors submitted for reconsideration. The decision letter after peer review and the authors’ response are shown below.]

Thank you for choosing to send your work entitled “Essential role of lncRNA binding for WDR5 maintenance of active chromatin and embryonic stem cell pluripotency” for consideration at *eLife*. Your full submission has been evaluated by a Senior editor, a Reviewing editor, and two peer reviewers, and this decision letter is based on discussions between the reviewers. We consider your work to be of potential importance but feel that you would have to make substantial changes to the manuscript before it could be acceptable for publication. We aim to publish articles with a single round of revision that would typically be accomplished within two months. We are not convinced that you will be able to deal with the criticisms during that time-frame but we are offering you the opportunity to inform us whether it is possible. If you are unable to revise the manuscript appropriately the manuscript will have to be rejected.

We would offer you the option of resubmitting if you can provide a reasonable response to the three major concerns below with a set of simple additional experiments that can be completed in a reasonable time frame.

You and your co-workers investigate WDR5, a protein previously demonstrated to bind the lncRNAs HOTTIP and NEST in differentiated cells. Using complementation to replace Wdr5 with wildtype or mutant human WDR5 in mESCs, you show that WDR5 F266A mutation in the lncRNA-binding pocket renders WDR5 unable to maintain stem cell pluripotency and global H3K4Me3 patterns. LncRNAs have been a highlight of post-genomic biology. This work is a timely contribution to understanding lncRNAs, considering that many lncRNA papers are descriptive expression surveys, with few interrogating lncRNA-protein interactions, let alone such interactions’ cellular functions.

1) However, you did not pursue WDR5 wildtype versus mutant interactions with known WDR5 lncRNA targets, such as HOTTIP, in cells where WDR5 co-occurs with its known targets, for example fibroblasts. Instead, you chose stem cells, which do not express WDR5-interacting lncRNAs. Although functional lncRNAs in pluripotency and differentiation are known (Sheik Mohamed et al., RNA 16: 324-337, 2010), you did not perform WDR5 RIPseq. You should attempt WDR5 RIP-qRTPCR to test for known ES lncRNAs as interactors, then confirm whether any interactions are mediated by the residues implicated in the HOTTIP binding.

2) The paper does not demonstrate a WDR5-lncRNA interaction essential to stemness. By the your own admission, “As HOTTIP and NeST RNAs are not expressed in ESCs, our results ... imply the existence of other ESC lncRNAs that stabilize WDR5.” Any resubmission should show rather than imply these lncRNAs’ existence, and test whether they interact with WDR5 in a manner abolished by F266A.

3) The effect of F266A is presented in terms of global H3K4Me3 signatures and expression level differences of multiple cellular state defining factors. This is potentially at odds with the target-specific lncRNA action model, where PRC2-interacting lncRNAs regulate well-defined sets of targets for each RNA, rather than cause global chromatin remodeling.

---

## [Author Response]

We thank the reviewers for the positive assessment and insightful critiques; your suggestions have improved the paper. Based on the reviewer comments, we undertook RNA immunoprecipitation experiments to define the set of RNAs bound by WDR5 in embryonic stem cells. The editors were right to guess that this effort took longer than the two months that we initially estimated. But we felt it was important to do the experiment right and be confident with the results (please see detailed information on approach, controls, and interpretation below). We are pleased to submit a revised version of the manuscript. If it is felt that this long revision process mandates consideration of the paper as a new submission, we can understand that decision as well.

*1) However, you did not pursue WDR5 wildtype versus mutant interactions with known WDR5 lncRNA targets, such as HOTTIP, in cells where WDR5 co-occurs with its known targets, for example fibroblasts. Instead, you chose stem cells, which do not express WDR5-interacting lncRNAs. Although functional lncRNAs in pluripotency and differentiation are known (Sheik Mohamed et al., RNA 16: 324-337, 2010), you did not perform WDR5 RIPseq. You should attempt WDR5 RIP-qRTPCR to test for known ES lncRNAs as interactors, then confirm whether any interactions are mediated by the residues implicated in the HOTTIP binding*.

We thank the reviewers for the suggestion, and agree that identifying endogenous targets of WDR5 in ESCs is important to understand its function. We tested UV-crosslinking with PAR- CLIP, but found that WDR5-RNA interactions have poor inherent UV crosslinking ability. We also found that standard RNA immunoprecipitation (RIP) with FLAG epitope gave substantial background that hampered data interpretation. We then turned to RNA:protein immunoprecipitation in tandem (RIPiT), a method designed to identify RNA targets of RBP complexes with poor UV linking capacity (Singh et al., Methods, 2013). Specifically, we fused tandem FLAG and hemagglutinin (HA) tags to wild type WDR5 or the F266A mutant, and generated stable ESC lines expressing either fusion protein at near endogenous levels (Figure 4—figure supplement 1). FPLC analysis confirmed that both tandem-tagged WDR5 proteins quantitatively formed equivalent MLL-WDR5 protein complexes with endogenous subunits (Figure 4—figure supplement 1).

We next performed RIPiT-seq of WDR5 in ESC with formaldehyde crosslinking and tandem IP. We identified over 1400 RNAs that bound WDR5, but importantly not to F266A mutant or vector control (revised Figure 4). Thus, a family of ESC RNAs bind WDR5 through the same surface as HOTTIP and NeST, enhancer-like RNAs that act in other cell types. WDR5 binds many types of RNAs including messenger RNAs, lncRNAs, and precursors of small regulatory RNAs, dramatically expanding the known RNA targets of this protein. Importantly, six lncRNA partners of WDR5 are known to be important for ESC pluripotency or differentiation based on published knockdown experiments (revised Figure 4). Intriguingly, prior protein interaction surveys with repressive chromatin modification complexes had failed to identify protein partners for these lncRNAs (Guttman et al., Nature, 2011), and their interaction with WDR5-MLL, a complex that enforces active chromatin, provides an explanation for their functions in ESC biology. The four lncRNAs identified by Sheik Mohamed et al. were not enriched in WDR5 RIPiT-seq.

We found a subset of WDR5-bound lncRNAs are encoded by loci that are also bound on chromatin by WDR5 in cis, while many other WDR5-bound lncRNAs reside in WDR5-free chromosomal loci, suggesting potential roles in trans (revised Figure 4). The chromatin binding profiles of WDR5 at cis-RNA bound loci vs RNA-independent loci are indistinguishable as a class, highlighting the value of the RIPiT data set to reveal RNA involvement (revised Figure 4). Collectively, the discovery of a family of ESC RNAs that bind WDR5 specifically through a RNA binding pocket will serve as a valuable resource for the field. The defect of the F266A mutant in binding specific lncRNAs required for ESC pluripotency provides a direct mechanistic links to its phenotypes in defective ESC self renewal and ectopic differentiation.

*2) The paper does not demonstrate a WDR5-lncRNA interaction essential to stemness. By the your own admission, “As HOTTIP and NeST RNAs are not expressed in ESCs, our results ... imply the existence of other ESC lncRNAs that stabilize WDR5.” Any resubmission should show rather than imply these lncRNAs’ existence, and test whether they interact with WDR5 in a manner abolished by F266A*.

As indicated in response to point 1 above, WDR5 RIPiT-seq revealed six lncRNA partners of WDR5 are known to be important for ESC pluripotency or differentiation based on published knockdown experiments (revised Figure 4). These lncRNAs bind WT WDR5 but not F266A mutant. Prior protein interaction surveys with repressive chromatin modification complexes had failed to identify partners for these lncRNAs (Guttman et al., Nature, 2011), and their interaction with WDR5-MLL, a complex that enforces active chromatin, provides an explanation for their functions in ESC biology. As an example, linc-1552 is a WDR5-interacting lncRNA that acts in a positive feedback circuit for pluripotency. Oct4 and Nanog bind linc-1552 promoter, and depletion of linc-1552 leads to loss of Nanog and Oct4 expression. This mechanism parallels the one described by Lipovich and colleagues (Sheik Mohamaed et al., RNA, 2010) for four lncRNAs they identified to be important for ESC stemness.

*3) The effect of F266A is presented in terms of global H3K4Me3 signatures and expression level differences of multiple cellular state defining factors. This is potentially at odds with the target-specific lncRNA action model, where PRC2-interacting lncRNAs regulate well-defined sets of targets for each RNA, rather than cause global chromatin remodeling*.

We thank the reviewers for recognizing this crucial point in our paper. The lncRNA field has focused on understanding the function of one lncRNA at a time, and we believe that a strength of our work is the elucidation of the cumulative effects of entire classes of lncRNAs. Using the above example where each PRC2-interacting lncRNA regulates a well-defined set of targets, the entire class of PRC2-interacting lncRNAs likely controls thousands of genes, either positively or negatively (Davidovich et al. and Kaneko et al., NSMB, 2013, cited in text). In this paper, we have elucidated the cumulative effects of the entire class of WDR5-binding RNAs in ES cells by replacing WDR5 WT with a RNA binding-null mutant, and we found that this single perturbation to WDR5-MLL complex results in global changes in H3K4me3 and cell fate, highlighting the importance of RNA input into WDR5.